# Guided Stream of Search: Learning to Better Search with Language Models via Optimal Path Guidance

## Abstract

While language models have demonstrated impressive capabilities across a range of tasks, they still struggle with tasks that require complex planning and reasoning. Recent studies have proposed training language models on search processes rather than optimal solutions, resulting in better generalization performance even though search processes are noisy and even suboptimal. However, these studies overlook the value of optimal solutions, which can serve as step-by-step landmarks to guide more effective search. In this work, we explore how to leverage optimal solutions to enhance the search and planning abilities of language models. To this end, we propose *guided stream of search* (GSoS), which seamlessly incorporates optimal solutions into the self-generation process in a progressive manner, producing high-quality search trajectories. These trajectories are then distilled into the pre-trained model via supervised fine-tuning. Our approach significantly enhances the search and planning abilities of language models on Countdown, a simple yet challenging mathematical reasoning task. Notably, combining our method with RL fine-tuning yields further improvements, whereas previous supervised fine-tuning methods do not benefit from RL. Furthermore, our approach exhibits greater effectiveness than leveraging optimal solutions in the form of subgoal rewards.

## 1 Introduction

Transformer-based language models have achieved remarkable success, demonstrating human-level performance across a wide range of natural language tasks, including conversation, code generation, and mathematical problem-solving (Achiam et al., 2023; Touvron et al., 2023; Roziere et al., 2023; Li et al., 2023; Lightman et al., 2023; Shao et al., 2024). Their impressive performance is primarily attributed to auto-regressive training on high-quality, internet-scale data. However, language models still face challenges with tasks that require complex planning and reasoning (Pallagani et al., 2023; Valmeekam et al., 2023). Models trained using next-token prediction often result in the snowballing of errors over long sequences, making it difficult for them to maintain consistent plans over multiple steps. Furthermore, teacher-forcing, where models are given the correct sequence of previous tokens for each prediction, exacerbates this problem by encouraging them to learn shortcuts rather than truly understanding the underlying structure of the task (Bachmann & Nagarajan, 2024).

A growing body of literature attempts to improve the planning and reasoning capabilities of language models through prompt-based strategies, allowing them to perform chain-of-thought reasoning, self-correction, and planning with symbolic search algorithms (Wei et al., 2022; Wang et al., 2023; Shinn et al., 2023; Yao et al., 2023). While these methods are successful in certain tasks, they have inherent limitations. They only assist the model during inference without updating its internal weights, which significantly constrains performance to that of the base model. Moreover, their success heavily relies on the quality of prompt design, and a poorly constructed prompt sometimes degrades performance (Huang et al., 2024).

To address these limitations, recent studies have shifted toward directly improving the planning and reasoning abilities of language models during the training phase (Lehnert et al., 2024; Gandhi et al., 2024). This approach, known as *stream of search* (SoS), involves training a model to predict search trajectories that encompass the entire decision-making process of finding solutions through trial and

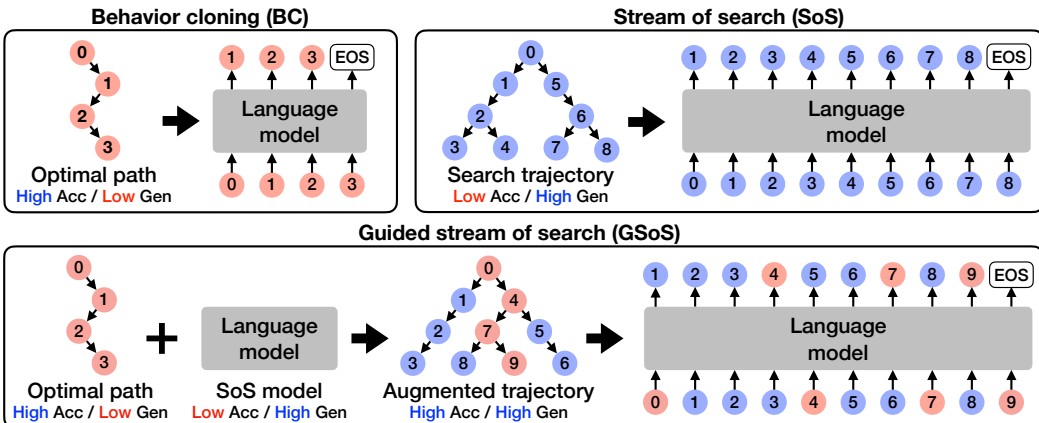

Figure 1: An overview of GSoS. It generates augmented trajectories by incorporating optimal paths into the self-generation process of the SoS model, achieving high accuracy and generalizability. The numbers indicate the order in which nodes are explored.

error, including exploration and backtracking when encountering failures. These studies have shown that models trained on search trajectories, despite being noisy and even suboptimal, achieve superior generalization performance to those trained to imitate optimal action sequences via *behavior cloning* (BC) (Ross et al., 2011). They have also demonstrated that fine-tuning on self-generated data further improves the search and planning capabilities of the models to some extent. However, this leads to a critical question: if models trained on search trajectories perform better, does this imply that optimal solutions are unnecessary? Optimal action sequences can serve as valuable landmarks, or subgoals, within the search process, guiding the model toward more effective search strategies. However, prior studies have underexplored the role of optimal solutions, focusing exclusively on fully self-generated data to improve the model, which is inherently limited.

In this work, we explore how optimal solutions can be leveraged to enhance the search and planning abilities of language models pre-trained on search trajectories. We observe that the pre-trained model successfully discovers solutions by searching when prompted with partial optimal solutions as hints, even when these solutions are unseen during pre-training. While this guidance produces high-quality data for supervised fine-tuning, it often has a low likelihood under the pre-trained model, potentially causing negative effects on the model's search and planning capabilities. To address this problem, we propose *guided stream of search* (GSoS), which seamlessly incorporates optimal solutions to direct the self-generation process and produces trajectories that have both high quality and likelihood. The key idea of GSoS is to integrate each intermediate action from the optimal solution into the trajectory in a step-by-step manner using unsuccessful search traces as contexts, as illustrated in Figure 1. We then fine-tune the pre-trained model to predict these augmented trajectories.

We evaluate our approach on Countdown (Gandhi et al., 2024), a mathematical reasoning benchmark that requires complex search and planning. Notably, our approach achieves an accuracy gain of 13% compared to the pre-trained model. When further applying RL fine-tuning tailored to handle the long contexts of search trajectories, this improvement increases to 20%, substantially outperforming both supervised and RL fine-tuned baselines. In contrast, standard supervised fine-tuning methods, which rely solely on self-generated data, do not benefit from RL fine-tuning. We conduct a comprehensive analysis of how unsuccessful search trajectories play an important role in generating trajectories with both high quality and likelihood. Moreover, we demonstrate that our approach is more effective than leveraging optimal solutions as subgoal rewards during RL fine-tuning (Lightman et al., 2023).

## 2 PRELIMINARIES

### 2.1 PROBLEM SETUP

In this paper, we consider a sequential decision-making problem derived from the program synthesis literature, which requires strong search and planning abilities (Devlin et al., 2017; Shi et al., 2024).

Each instance of this problem is defined as a tuple $x = (I, O)$, where $I$ denotes the input state and $O$ denotes the output state. We are given a set of operations $\mathcal{P}$, where each operation $P \in \mathcal{P}$ maps one state to another, producing an intermediate state. There exists an optimal sequence of operations $\hat{y} = (P_1, P_2, \ldots, P_N)$ that transforms the input state into the output state, i.e., $\hat{y} = P_N \circ \cdots \circ P_2 \circ P_1(x)$. Given a training set $\mathcal{D}$, our goal is to train a model $f_\theta$ to generate optimal solutions for a test set.

## 2.2 STREAM OF SEARCH

One straightforward approach to this problem is to apply imitation learning through BC (Ross et al., 2011), which trains the model to directly predict optimal solutions from input-output pairs. However, prior studies have demonstrated that BC struggles to generalize to unseen test examples (Yang et al., 2022; Lehnert et al., 2024; Gandhi et al., 2024).

To address this, SoS introduces an approach that leverages the search process rather than the optimal solution (Gandhi et al., 2024). This method reformulates the problem as a tree search, navigating the search tree through trial and error with operations until the solution is reached. Each node in the tree represents a state, and each edge represents an operation between two states. SoS expresses primitive operations for the tree search in language, including node generation, exploration, backtracking, and verification. It then generates search trajectories using symbolic search algorithms, including depth-first search (DFS) and breadth-first search (BFS), representing them as sequences of tokens. Finally, it trains a language model to predict these trajectories from input-output pairs used as prompts:

$$\max_\theta \mathbb{E}_{x \sim \mathcal{D}, y \sim \text{symbolic}(x)} \left[ \log f_\theta(y \mid x) \right].$$

Note that the SoS model is not specifically designed to find solutions, as symbolic search algorithms may produces suboptimal trajectories with a limited search budget. See Appendix A for more details.

## 2.3 SUPERVISED FINE-TUNING

The pre-trained language model, not originally designed for particular tasks, requires alignment with a downstream task to enhance its performance. One approach to achieving this alignment is to apply supervised fine-tuning with self-generated data, referred to as self-taught reasoner (STaR) (Zelikman et al., 2022; Gulcehre et al., 2023). This method generates trajectories using the model from prompts and filters them based on a task-specific metric $M$ that evaluates quality. It then fine-tunes the model to predict these filtered trajectories:

$$\max_\theta \mathbb{E}_{x \sim \mathcal{D}, y \sim f_\theta(\cdot \mid x)} \left[ \mathbb{1}\left[ M(y \mid x) > \tau \right] \cdot \log f_\theta(y \mid x) \right], \tag{1}$$

where $\tau$ is the threshold that controls the ratio of filtered trajectories. This process can be performed iteratively to further refine the model.

## 2.4 REINFORCEMENT LEARNING FINE-TUNING

Another approach to aligning the pre-trained language model is to perform RL fine-tuning (Stiennon et al., 2020; Ouyang et al., 2022). This method uses the model as a policy $\pi_\theta$ and fine-tunes the policy to maximize a task-specific reward $R$, while minimizing the KL divergence from the reference policy $\pi_{\text{ref}}$. The problem is formulated as a token-level finite-horizon Markov decision process (MDP). The state space $\mathcal{S}$ is the set of all possible token sequences, and the action space $\mathcal{A}$ is the set of all tokens. The initial state $s_0 \in \mathcal{D}$ is a randomly sampled prompt. Each state $s_h \in \mathcal{S}$ is the concatenation of the prompts and the previously generated tokens, with the transition function $p : \mathcal{S} \times \mathcal{A} \to \mathcal{S}$ appending the action $a_h \in \mathcal{A}$ to the state. Given a trajectory generated by the policy with horizon $H \in \mathbb{Z}^+$, the policy is trained to optimize the following objective:

$$\max_\theta \mathbb{E}_{s_0 \sim \mathcal{D}, (s_h, a_h) \sim \pi_\theta(\cdot \mid s_0)} \left[ \sum_{h=0}^{H} R(s_h, a_h) - \beta \cdot (\log \pi_\theta(a_h \mid s_h) - \log \pi_{\text{ref}}(a_h \mid s_h)) \right],$$

where $\beta > 0$ is the coefficient that controls the influence of the KL divergence. Typically, the reward is provided at the end of the trajectory.

A common algorithm for training this policy is proximal policy optimization (PPO) (Schulman et al., 2017). This method optimizes the policy using the clipped surrogate objective, which is designed to limit policy changes. Simultaneously, it trains a value function $V_\phi : \mathcal{S} \to \mathbb{R}$ to predict the multi-step returns computed by the generalized advantage estimator (GAE) (Schulman et al., 2016).

## 2.5 COUNTDOWN BENCHMARK

To evaluate the search and planning abilities of language models, we use Countdown as a benchmark (Gandhi et al., 2024). Each problem consists of input numbers and a target number, where the goal is to transform the inputs into the target using the four basic arithmetic operations. This problem has a high branching factor of $O(k^2)$ in the search tree, where $k$ is the number of remaining inputs. To ensure a tractable level of difficulty, we set the number of initial inputs to 4, following the setup in Gandhi et al. (2024). See Appendix B for more details on Countdown.

## 3 METHODS

### 3.1 MOTIVATION

Gandhi et al. (2024) show that training a language model using noisy, suboptimal search trajectories leads to better generalization compared to clean, optimal solutions. That said, optimal solutions can still offer valuable guidance during generation. If the SoS model has the ability to continue the search from arbitrary incomplete trajectories, we can leverage this to generate high-quality trajectories. By providing the model with partial optimal solutions alongside the prompts as hints, we can guide it to continue the search within a reduced search space, increasing the probability of finding a solution.

To examine whether the SoS model possesses this ability, we conduct an experiment on Countdown. For each problem with an optimal solution of $N$ operations, we define the partial optimal path as the trajectory generated by applying the first $n$ operations in the search tree. We then append this path to the initial prompt. Finally, we generate search trajectories using the SoS model from these modified prompts and evaluate correctness. Table 1 shows the ratio of successful trajectories with varying the length of partial optimal solutions for 200,000 training problems. The model successfully discovers solutions starting from these paths, despite not having encountered them during training. Moreover, increasing their length greatly improves the correctness of the resulting trajectories. This encourages us to use these high-quality, self-generated data for supervised fine-tuning, distilling the knowledge of optimal solutions into the model.

However, this guidance results in low likelihood under the model. Table 1 presents the cross-entropy loss of the trajectories, showing that using longer partial optimal paths results in higher loss values. Fine-tuning on these trajectories may lead to significant changes in the model's weights, potentially degrading its search and planning abilities. Therefore, it is crucial to explore methods for effectively integrating optimal solutions to produce trajectories that maintain both high likelihood and quality.

Table 1: Ratio and loss of successful SoS trajectories with varying the length of partial optimal paths. "Length 0" represents no paths are appended, and "Length 3" represents full paths are appended.

| Length | 0 | 1 | 2 | 3 |
|---|---|---|---|---|
| Ratio (↑) | 0.4918 | 0.7281 | 0.9211 | **1.0000** |
| Loss (↓) | **0.0742** | 0.1133 | 0.2301 | 0.3509 |

### 3.2 GUIDED STREAM OF SEARCH

In this subsection, we introduce *guided stream of search* (GSoS), a supervised fine-tuning approach that seamlessly incorporates optimal solutions into the self-generation process and effectively distills them into the model. The key idea of GSoS is to leverage an unsuccessful search trajectory as context for each intermediate step of the optimal solution. This approach effectively mimics how the model discovers the solution through the search procedure, producing a trajectory that has a high likelihood under the SoS model. Moreover, by providing an exploratory context for reaching each intermediate step, it facilitates distilling the optimal solutions into the model (Yang et al., 2022).

Before delving into our approach, we establish some notations. We define subgoal nodes as the non-leaf nodes along the optimal path in the search tree. An optimal solution consisting of $N$ operations contains $N-1$ subgoal nodes. We define the generation and exploration lines of a node as sequences of tokens that represent the primitive operations for generating and exploring the node, respectively.

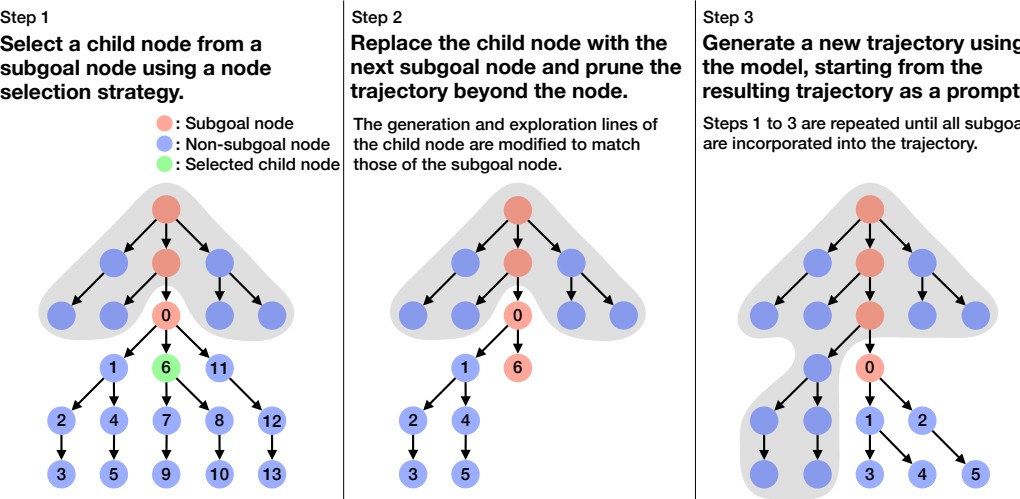

Figure 2: An illustration of subgoal augmentation in GSoS. The numbers indicate the order in which nodes are explored.

Note that while we include the root node as a subgoal node for simplicity, it is always explored prior to the search since it is already present in the prompt.

GSoS incorporates each subgoal into a trajectory in a progressive manner. Consider an unsuccessful search trajectory that has reached the $(n-1)$-th subgoal but fails to navigate its subtree, thereby not reaching the $n$-th subgoal. First, it selects the $k$-th explored child node from the $(n-1)$-th subgoal node. Subsequently, it replaces this child node with the $n$-th subgoal node, modifying its generation and exploration lines within the trajectory. While multiple exploration lines may exist, only the first one is modified. Finally, it prunes the trajectory beyond the modified exploration line, allowing the model to restart the search from the subgoal node. An example of the resulting trajectory is provided in Figure 12 in Appendix C. GSoS uses this trajectory as a prompt to generate a new search trajectory using the SoS model. This procedure, called *subgoal augmentation*, is summarized in Figure 2 and Algorithm 1. It is repeated until all subgoals are incorporated into the trajectory.

---

**Algorithm 1** Subgoal augmentation

---

**Require:** model $f_\theta$, search trajectory $y$, optimal path $\hat{y}$, subgoal index $n$
1: **if** $y$ has explored the $n$-th subgoal node of $\hat{y}$ **then**
2:     **return** $y$
3: **end if**
4: Select an explored child node from the $(n-1)$-th subgoal node of $\hat{y}$
5: Replace the child node with the $n$-th subgoal of $\hat{y}$
6: Prune $y$ starting from the node
7: Generate a new search trajectory: $y \leftarrow f_\theta(y)$
8: **return** $y$

---

Note that the exploration history from the first to the $(k-1)$-th explored child node is used as context for the subgoal node, where $k$ determines the size of this context. We consider three node selection strategies to control the context size and use random node selection as the base strategy:

- **First**: This strategy selects the first explored child node, providing minimal context.
- **Rand**: This strategy randomly selects an explored child node, providing varied context.
- **Last**: This strategy selects the last explored child node, providing maximal context.

Finally, GSoS filters these augmented trajectories using a task-specific metric and fine-tunes the SoS model on the filtered data, resulting in the GSoS model. Algorithm 2 outlines the overall procedure of GSoS, encompassing both subgoal augmentation and supervised fine-tuning.

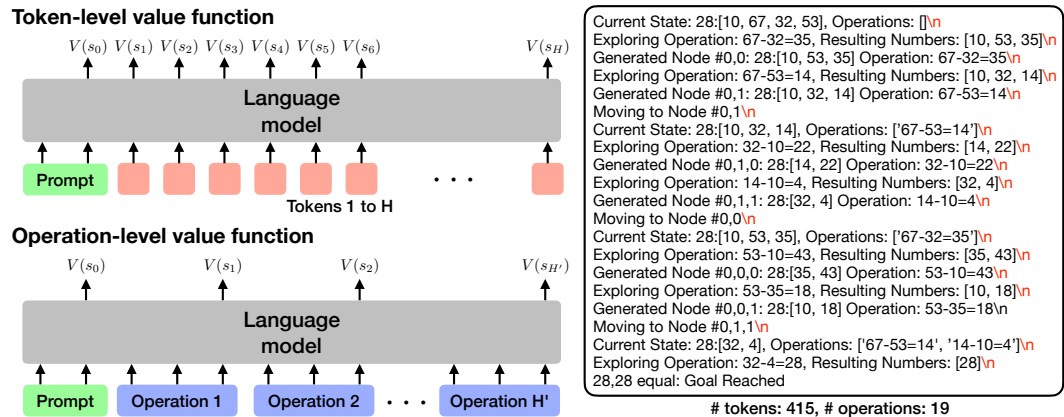

Figure 3: **(Left)** Illustration of token-level and operation-level value functions. **(Right)** An example of a search trajectory with values computed at newline tokens in the operation-level MDP.

---

**Algorithm 2** Guided stream of search

---
**Require:** SoS model $f_\theta$, prompt $x$, optimal path $\hat{y}$, task-specific metric $M$, threshold $\tau$
1: Initialize the model: $f_{\theta_1} \leftarrow f_\theta$
2: **for** $i = 1, 2, \ldots, \text{MAXITER}$ **do**
3:     Generate a search trajectory: $y \leftarrow f_{\theta_i}(x)$
4:     **for** $n = 1, 2, \ldots, N - 1$ **do**
5:         **if** $M(y \mid x) > \tau$ **then**
6:             **break**
7:         **end if**
8:         Augment the $n$-th subgoal with Algorithm 1: $y \leftarrow \text{AUGMENTSUBGOAL}(f_{\theta_i}, y, \hat{y}, n)$
9:     **end for**
10:     Train the model with Equation (1): $f_{\theta_{i+1}} \leftarrow \text{TRAIN}(f_\theta, x, y, M, \tau)$
11: **end for**

---

### 3.3 OPERATION-LEVEL RL FINE-TUNING

Building on the GSoS model, we apply RL fine-tuning to further enhance its performance. However, this poses challenges since the model generates search trajectories that are much longer than optimal paths. The extremely long sequences, combined with the sparse rewards given only at the end, hinder the effective propagation of the reward signal to earlier timesteps, even when multi-step returns with GAE are used for training the value function.

To address this, we formulate the problem as an operation-level MDP, which significantly reduces its effective horizon. The action space consists of the primitive operations for tree search defined in SoS, separated by newline tokens within the trajectory, as shown in Figure 3. The state space includes all possible operation sequences ending with a newline token. Accordingly, the value function is trained to predict values exclusively at these newline tokens, as illustrated in Figure 3. This operation-level approach reduces the effective horizon to less than 5%, facilitating better reward signal propagation. See Appendix D for a more detailed explanation.

## 4 EXPERIMENTS

### 4.1 EXPERIMENT SETUP

Throughout all experiments, we use a GPT-2 architecture with 250M parameters and a context length of 4096 as the base language model (Radford et al., 2019). Additionally, we utilize FlashAttention-2 for faster training and inference (Dao, 2024). Although Gandhi et al. (2024) employ GPT-Neo as the base architecture (Gao et al., 2020), we find that using GPT-2 instead results in better performance. See Appendix E.2 for a more detailed comparison.

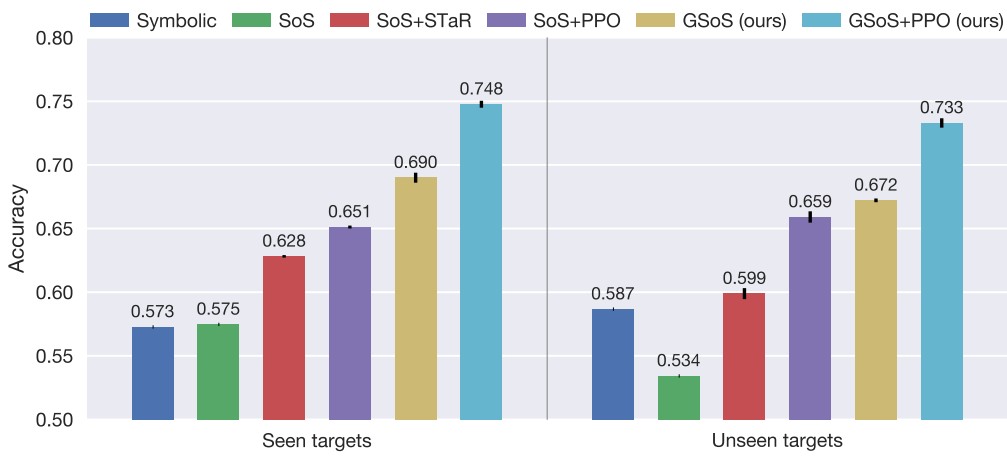

Figure 4: Test accuracy of each method. Each fine-tuning method is evaluated over three runs with different seeds. Error bars represent the standard deviation relative to the mean.

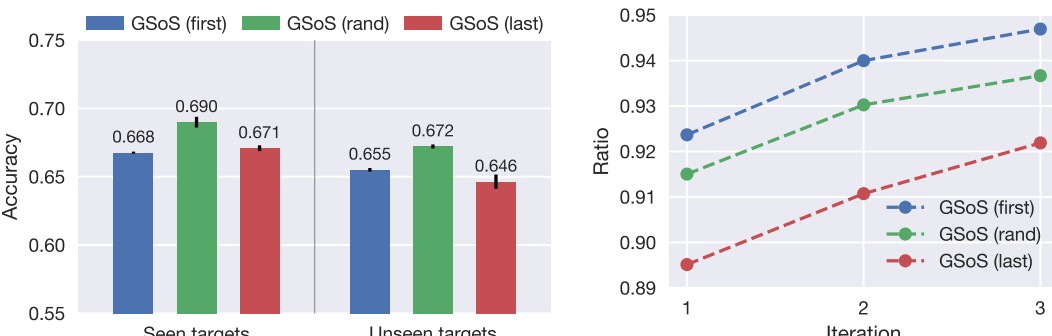

Figure 5: Test accuracy of GSoS with different node selection strategies.

Figure 6: Ratio of successful GSoS trajectories with different node selection strategies.

For unsupervised pre-training, we create 500,000 training problems and generate search trajectories using heuristic-guided DFS and BFS. For supervised fine-tuning, we generate trajectories from the first 200,000 training problems with a temperature of 0.8, filtering them based on their correctness. We perform this fine-tuning for three iterations, following Gandhi et al. (2024). For RL fine-tuning, we generate trajectories from 25,600 randomly sampled training problems with a temperature of 1.0. We use the same reward function as Gandhi et al. (2024), which takes into account both correctness and efficiency. See Appendices E.1 to E.4 for more details.

For evaluation, we follow the procedure outlined in Gandhi et al. (2024). We create 10,000 problems for each of the two test cases: (1) seen targets and (2) unseen targets. We generate trajectories using greedy decoding and measure accuracy based on their correctness. To evaluate models fine-tuned on self-generated data, which involves randomness in the data generation process, we run experiments with three different seeds and measure the mean and standard deviation. See Appendix E.1 for more details on the test set.

We compare our methods with the symbolic method, SoS, and the two baseline fine-tuning methods introduced in Gandhi et al. (2024): (1) SoS+STaR and (2) SoS+PPO. Although Gandhi et al. (2024) employ iterative APA as the base RL algorithm, but this approach necessitates resetting the reference policy, which introduces additional costs for model selection (Zhu et al., 2023). We find that a single iteration of PPO outperforms iterative APA. See Appendix F for a more detailed comparison. Unless otherwise specified, we use operation-level PPO for all RL fine-tuning experiments.

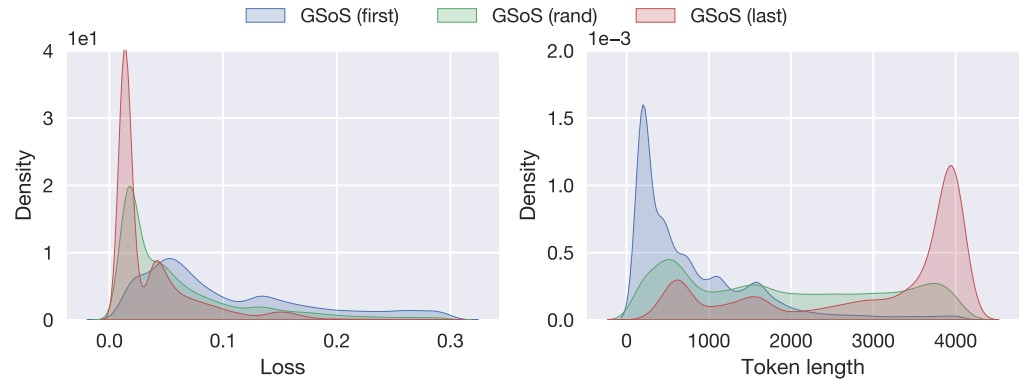

Figure 7: Distributions of losses (**left**) and token lengths (**right**) for successful GSoS trajectories at the final iteration with different node selection strategies.

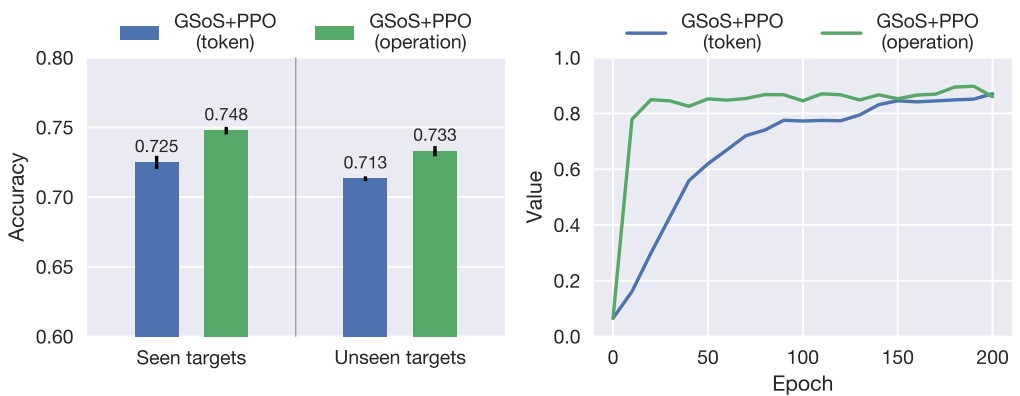

Figure 8: (**Left**) Test accuracy of GSoS+PPO in the token-level and operation-level MDPs. (**Right**) Value predictions for initial states of GSoS+PPO in the token-level and operation-level MDPs.

## 4.2 RESULTS

Figure 4 presents the test accuracy of each method for both seen and unseen targets. Our approaches outperform the symbolic method and SoS by a substantial margin. Specifically, GSoS+PPO achieves 75% accuracy on seen targets and 73% accuracy on unseen targets, yielding an average gain of over 16% compared to both methods. Furthermore, our approaches significantly outperform the baseline fine-tuning methods. For supervised fine-tuning, GSoS shows a gain of 7% compared to SoS+STaR. For RL fine-tuning, GSoS+PPO exhibits a gain of 9% compared to SoS+PPO. It is worth noting that RL fine-tuning further enhances the performance of GSoS by an additional 6%.

## 4.3 ANALYSIS OF GSOS NODE SELECTION STRATEGIES

Figure 5 shows the test accuracy of GSoS with different node selection strategies. The random node selection strategy achieves the highest accuracy for both target types, outperforming the second-best strategy by 2%. To investigate the effectiveness of random node selection, we analyze the statistics of trajectories generated by each strategy. Figure 7 displays the distributions of cross-entropy losses for successful GSoS trajectories at the final iteration, where the losses are calculated on the base SoS model. Strategies with longer contexts generate trajectories with higher likelihoods under the model compared to shorter contexts. However, this benefit comes at the cost of lower quality. As shown in Figure 6, the ratio of successful trajectories decreases as the context length increases. One potential reason is that longer contexts reduce the number of tokens available for the model to generate within a given context limit, thereby restricting its opportunities to search. This explanation is supported by the fact that many of the successful trajectories generated by the last node selection strategy already approach the context length limit, as presented in Figure 7. In summary, there is an inherent trade-off

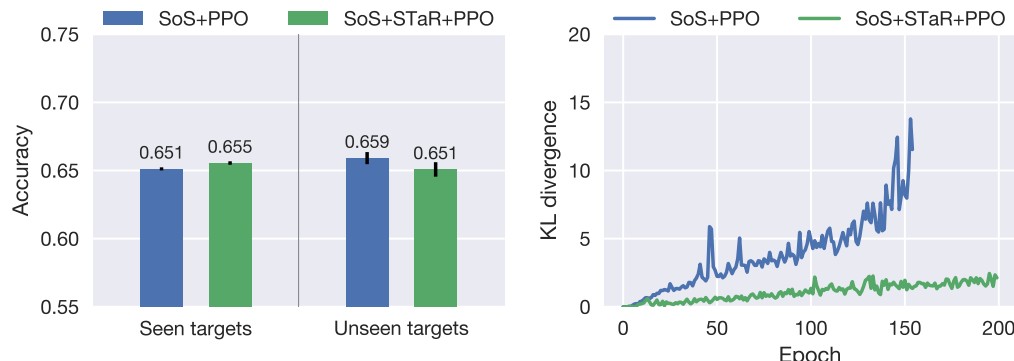

Figure 9: (**Left**) Test accuracy of SoS+PPO with and without STaR. (**Right**) KL divergence from the reference policy of SoS+PPO with and without STaR. SoS+PPO terminates early due to instability.

between quality and likelihood. Nevertheless, the random node selection strategy strikes a balance between this trade-off.

## 4.4    ANALYSIS OF OPERATION-LEVEL PPO

To examine the effectiveness of the operation-level action space for RL fine-tuning, we fine-tune the GSoS model using PPO in the token-level MDP and compare their performance. Figure 8 shows the test accuracy and value predictions for initial states of GSoS+PPO in the token-level and operation-level MDPs. Notably, the operation-level approach achieves an accuracy gain of over 2% compared to the token-level approach for both target types. Moreover, the operation-level value function learns at a significantly faster rate than the token-level value-function, highlighting the benefits of reducing the effective horizon for training the value function.

## 4.5    ANALYSIS OF RL FINE-TUNING ON STAR

In Section 4.2, we find that RL fine-tuning further enhances the performance of GSoS. This raises the question of whether RL fine-tuning might also benefit SoS+STaR. To explore this, we fine-tune the SoS+STaR model using PPO and compare their performance. Figure 9 shows the test accuracy and KL divergence from the reference policy of SoS+PPO with and without STaR. While applying STaR prior to RL fine-tuning leads to lower KL divergence and improved training stability, its performance remains similar. In summary, STaR does not provide additional benefits beyond RL fine-tuning. This highlights the importance of optimal solutions in enhancing performance.

Table 2: Test accuracy of SoS+PPO with and without subgoal reward.

| Model | Accuracy (seen) | Accuracy (unseen) |
|---|---|---|
| SoS+PPO | $0.6512 \pm 0.0011$ | $0.6591 \pm 0.0044$ |
| SoS+PPO (subgoal) | $0.6597 \pm 0.0011$ | $0.6650 \pm 0.0018$ |
| GSoS+PPO | $\mathbf{0.7477 \pm 0.0027}$ | $\mathbf{0.7330 \pm 0.0037}$ |

## 4.6    ANALYSIS OF RL FINE-TUNING WITH SUBGOAL REWARD

Another approach to distilling the information from optimal solutions into the SoS model is to apply RL fine-tuning with a subgoal reward. Specifically, we train the model using PPO with a new reward function that combines the original reward and a subgoal reward, which is defined as:

$$R_{\text{new}}(s_t, a_t) = R(s_t, a_t) + \eta \cdot R_{\text{sub}}(s_t, a_t), \quad R_{\text{sub}}(s_t, a_t) = \begin{cases} 1 & \text{if } (s_t, a_t) \text{ explores a subgoal} \\ 0 & \text{otherwise,} \end{cases}$$

where $\eta > 0$ controls the influence of the subgoal reward. We set $\eta$ to 0.2 to ensure that the subgoal reward does not dominate the original reward. To avoid exploitation, the subgoal reward is provided

only for the initial exploration, which is motivated by Hafner (2022). Table 2 shows the test accuracy of SoS+PPO with and without the subgoal reward. Adding the subgoal reward leads to only marginal improvements, showing an accuracy gain of less than 1% for both target types. This underscores the critical role of GSoS for distilling optimal solutions to the model.

## 5 RELATED WORKS

**Searching with language models**  Yang et al. (2022) introduce the idea of using search procedure for sequential decision making. They train a neural network to imitate not only actions generated by an expert MCTS policy but also the search procedures involved in determining the actions. However, their approach primarily focuses on utilizing search procedures to enhance imitation learning rather than improving the search and planning capabilities of the model. Recent studies focus on enhancing these capabilities using language models. Lehnert et al. (2024) train a T5 model to imitate $A^*$ search traces and optimize it to generate shorter traces through supervised fine-tuning (Raffel et al., 2020). Gandhi et al. (2024) train a GPT-Neo model to imitate DFS and BFS search trajectories and improve its performance through supervised or RL fine-tuning. Our work differs from these prior studies in that it utilizes optimal solutions throughout the self-data generation process rather than relying solely on fully self-generated data.

**Fine-tuning on self-generated data**  Anthony et al. (2017); Silver et al. (2017) introduce the idea of iteratively distilling self-generated data into neural networks to enhance their performance. They generate high-quality trajectories using an expert MCTS policy and imitate them to improve a neural network-based policy. However, their work is restricted to small-scale convolutional neural networks and the narrow domain of gaming. Recent studies extend this idea to fine-tune language models for various downstream tasks, including theorem proving, question answering, and machine translation (Polu et al., 2022; Zelikman et al., 2022; Gulcehre et al., 2023). While Zelikman et al. (2022) utilize optimal solutions as prompts to generate higher-quality data, our approach differs by first generating trajectories without any prior knowledge of optimal solutions and then integrating optimal solutions into them. Furthermore, we incorporate each intermediate subgoal of optimal solutions step by step, in the same spirit as Lightman et al. (2023).

**RL fine-tuning with higher-level MDP**  Verma et al. (2022); Zhou et al. (2024) address the long-horizon problem in multi-turn conversation tasks by defining each single-turn utterance as an action. However, their work focuses exclusively on off-policy RL algorithms. Furthermore, their approaches require an additional value network for baseline estimation and a target network to improve training stability. Our work differs from these prior studies by successfully extending the idea of higher-level MDP to on-policy RL algorithms without introducing any additional networks. Although Ahmadian et al. (2024) explore sequence-level optimization using REINFORCE (Williams, 1992), their focus is limited to tasks with short horizons, where the average generation length is under 100.

## 6 CONCLUSION

In this work, we explore the role of optimal solutions in enhancing the search and planning abilities of language models. We identify that optimal solutions can serve as step-by-step subgoals, providing valuable guidance to models pre-trained on search trajectories. We propose *guided stream of search* (GSoS), which seamlessly integrates optimal solutions into the self-generation process step by step, producing high-quality trajectories for supervised fine-tuning. Our method achieves state-of-the-art performance on the challenging Countdown benchmark. Furthermore, we find that GSoS possesses the intriguing property of working in tandem with RL fine-tuning, which is not observed in standard supervised fine-tuning.

Our work is not without limitations. Transformer models trained on search trajectories face inherent challenges as they require long context lengths. Given their quadratic memory and time complexity, this significantly raises the computational demands for both training and inference. This issue can be further exacerbated as tasks grow more complex and require even longer context lengths. Exploring the search and planning capabilities of more efficient language models, such as Mamba (Gu & Dao, 2023), is a promising direction for future work. Alternatively, training a model to implicitly perform search through knowledge distillation also presents a valuable approach (Deng et al., 2023).

## REPRODUCIBILITY STATEMENT

We provide implementation details in Appendix E, including model configuration, hyperparameters for both training and inference, and the modifications made to the code repositories we used.

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

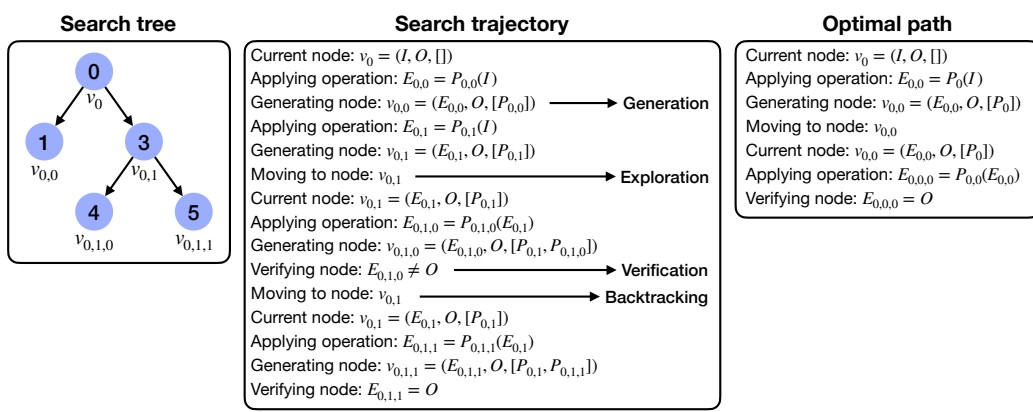

Figure 10: An example of a search tree, its corresponding search trajectory, and an optimal path in SoS.

# A  STREAM OF SEARCH

SoS trains a language model to predict search trajectories generated by symbolic search algorithms (Gandhi et al., 2024). For each problem with the input state $I$ and the output state $O$, its search tree is structured with the root node representing the input state, non-root nodes representing intermediate states, and edges representing operations between nodes. A leaf node is reached when no additional operations can be applied. Formally, each nodes and edge in the tree are defined as follows:

- **Node**: Each node $v_\alpha$ at depth $d$, where $\alpha = (i_1, i_2, \ldots, i_d)$, represents a sequence of $d - 1$ operations and the resulting intermediate states, along with the output state.
- **Edge**: Each edge $e_{v,v'}$ represents an operation that maps the node $v$ to the node $v'$.

SoS defines the following primitive operations for symbolic search algorithms in langauge:

- **Generation**: Given a node $v_\alpha$ at depth $d$, this action generates the $i_{d+1}$-th child node $v_{\alpha'}$ at depth $d + 1$ by applying an operation, where $\alpha' = (\alpha, i_{d+1})$.
- **Exploration**: Given a node $v_\alpha$ at depth $d$, this action transitions to the previously generated $i_{d+1}$-th child node $v_{\alpha'}$ at depth $d + 1$, where $\alpha' = (\alpha, i_{d+1})$.
- **Backtracking**: Given a node $v_\alpha$ at depth $d$, if exploring its child nodes is unlikely to yield a solution, this action transitions to a previously generated node $v_{\alpha'}$ at depth $d'$.
- **Verification**: Given a leaf node $v_\alpha$ at depth $D$, where no further operations can be applied, this action verifies whether the node represents the solution.

Finally, it generates search trajectories using diverse symbolic search algorithms. Since all primitive operations are expressed in language, search trajectories are also represented in language. Figure 10 illustrates an example of a search tree and its corresponding search trajectory. Note that an optimal solution can be represented as a path in a search tree.

One major challenge in training language models on these search trajectories is their context length. Symbolic search algorithms can generate excessively long search trajectories via exhaustive search. However, language models, such as Transformers (Vaswani et al., 2017), have a fixed context length, preventing them from capturing or learning from sequences beyond this length. To address this, SoS uses heuristic-guided search algorithms to generate shorter search trajectories, albeit with reduction in performance. See Appendix E.1 for more details on these algorithms.

# B  COUNTDOWN BENCHMARK

Each problem begins with $K$ input numbers and a target number, all of which are integers. The input numbers are either 1-digit or 2-digit, whereas the target number is 2-digit. Arithmetic operations are

---

**Symbolic search trajectory**

```
1   Current State: 25:[56, 58, 15, 8], Operations: []
2   Exploring Operation: 58-56=2, Resulting Numbers: [15, 8, 2]
3   Generated Node #0,0: 25:[15, 8, 2] Operation: 58-56=2
4   Moving to Node #0,0
5   Current State: 25:[15, 8, 2], Operations: ['58-56=2']
6   Exploring Operation: 8*2=16, Resulting Numbers: [15, 16]
7   Generated Node #0,0,0: 25:[15, 16] Operation: 8*2=16
8   Moving to Node #0,0,0
9   Current State: 25:[15, 16], Operations: ['58-56=2', '8*2=16']
10  Exploring Operation: 15+16=31, Resulting Numbers: [31]
11  31,25 unequal: No Solution
12  Moving to Node #0,0,0
13  Current State: 25:[15, 16], Operations: ['58-56=2', '8*2=16']
14  Exploring Operation: 16-15=1, Resulting Numbers: [1]
15  1,25 unequal: No Solution
16  Moving to Node #0,0
17  Current State: 25:[15, 8, 2], Operations: ['58-56=2']
18  Exploring Operation: 15*2=30, Resulting Numbers: [8, 30]
19  Generated Node #0,0,1: 25:[8, 30] Operation: 15*2=30
20  Moving to Node #0,0,1
21  Current State: 25:[8, 30], Operations: ['58-56=2', '15*2=30']
22  Exploring Operation: 30-8=22, Resulting Numbers: [22]
23  22,25 unequal: No Solution
24  Moving to Node #0,0,1
25  Current State: 25:[8, 30], Operations: ['58-56=2', '15*2=30']
26  Exploring Operation: 8+30=38, Resulting Numbers: [38]
27  38,25 unequal: No Solution
28  Moving to Node #0,0
29  Current State: 25:[15, 8, 2], Operations: ['58-56=2']
30  Exploring Operation: 15+8=23, Resulting Numbers: [2, 23]
31  Generated Node #0,0,2: 25:[2, 23] Operation: 15+8=23
32  Moving to Node #0,0,2
33  Current State: 25:[2, 23], Operations: ['58-56=2', '15+8=23']
34  Exploring Operation: 2+23=25, Resulting Numbers: [25]
35  25,25 equal: Goal Reached
```

Figure 11: An example of a search trajectory generated by a symbolic algorithm on Countdown.

limited to those yielding non-negative integers. Specifically, subtraction is restricted to cases where the larger number is subtracted from the smaller number. Similarly, division is only permitted when there is no remainder. Since all operations are binary, the number of inputs decreases by 1 after each operation, resulting in a search tree of depth $K - 1$. Figure 11 illustrates an example of a Countdown problem and its search trajectory generated by a symbolic search algorithm.

## C  GUIDED STREAM OF SEARCH

Figure 12 illustrates a trajectory generated by subgoal augmentation in GSoS on Countdown. Given the root node, the child node of index (0, 1) is selected and replaced with the second subgoal node. This involves modifying the operation and resulting numbers within the generation and exploration lines of the child node to match those of the subgoal node. Finally, all lines beyond the exploration line are truncated. The resulting trajectory is fed into the model to restart the search from the subgoal node. This process is repeated if the model fails to explore the next subgoal.

Figure 12: An example of a trajectory generated by subgoal augmentation in GSoS on Countdown. It modifies the generation and exploration lines of a selected child node to match those of the subgoal node, resulting in a new node.

## D  OPERATION-LEVEL RL FINE-TUNING

Before delving into operation-level RL fine-tuning, we first explain how RL fine-tuning is performed using PPO in the token-level MDP. Starting from the initial state $s_0$, the policy generates a trajectory $(s_0, a_0, s_1, a_1, \ldots, s_H)$ up to the horizon $H$ by sampling a token as an action and appending it to the current state at each timestep. The policy $\pi_\theta$ and value function $V_\phi$ are then trained on this trajectory using the following objectives:

$$\max_\theta \ \mathbb{E}_{s_0 \sim \mathcal{D}, (s_h, a_h) \sim \pi_\theta} \left[ \min \left( \frac{\pi_\theta(a_h \mid s_h)}{\pi_{\theta_{\text{old}}}(a_h \mid s_h)} \hat{A}_h, \text{clip} \left( \frac{\pi_\theta(a_h \mid s_h)}{\pi_{\theta_{\text{old}}}(a_h \mid s_h)}, 1 - \epsilon, 1 + \epsilon \right) \hat{A}_h \right) \right], \quad (2)$$

$$\min_\phi \ \mathbb{E}_{s_0 \sim \mathcal{D}, (s_h, a_h) \sim \pi_\theta} \left[ \frac{1}{2} \left( V_\phi(s_h) - \left( \hat{A}_t + V_{\phi_{\text{old}}}(s_h) \right) \right)^2 \right]. \quad (3)$$

Here, $\pi_{\theta_{\text{old}}}$ and $V_{\phi_{\text{old}}}$ are the policy and value function immediately prior to the update, and $\hat{A}$ is the advantage compute by GAE, which is defined as

$$\hat{A}_h = \sum_{h'=h}^{H} (\gamma\lambda)^l \delta_{h'}^V, \quad \delta_h^V = r_h + \gamma V_\phi(s_{h+1}) - V_\phi(s_h). \quad (4)$$

In the operation-level MDP, each action $a_h$ is defined as a sequence of tokens $(a_{h,1}, a_{h,2}, \ldots, a_{h,T})$, with the last token being a newline token. Consequently, each state $s_h$ is defined as the concatenation of the initial state and previously generated actions, with the last token being a newline token. In this MDP, calculating the advantage and multi-step return is straightforward: simply forward the state to the value function and apply Equation (4). The action probability is calculated by factorizing it over the sequence of tokens:

$$\pi_\theta(a_h \mid s_h) = \prod_{t=1}^{T} \pi_\theta(a_{h,t} \mid a_{h,t-1}, \ldots, a_{h,1}, s_h).$$

Finally, the policy and value function are trained to optimize Equations (2) and (3).

## E  IMPLEMENTATION DETAILS

### E.1  DATASET

We construct the dataset following the procedure outlined in Gandhi et al. (2024). First, we split the set of target numbers into 90% for training and 10% for testing. We create 500,000 training problems

```
1   Current State: 22:[35, 15, 66, 61], Operations: []
2   Exploring Operation: 66-61=5, Resulting Numbers: [35, 15, 5]
3   Generated Node #0,0: 22:[35, 15, 5] Operation: 66-61=5
4   Moving to Node #0,0
5   Current State: 22:[35, 15, 5], Operations: ['66-61=5']
6   Exploring Operation: 35-15=20, Resulting Numbers: [5, 20]
7   Generated Node #0,0,0: 22:[5, 20] Operation: 35-15=20
8   Moving to Node #0,0,0
9   Current State: 22:[5, 20], Operations: ['66-61=5', '35-15=20']
10  Exploring Operation: 20/5=4, Resulting Numbers: [4]
11  4,22 unequal: No Solution
12  Moving to Node #0,0,0,0
13  Current State: 22:[4], Operations: ['66-61=5', '35-15=20', '20/5=4']
14  No solution found.
```

**Incorrect BFS trajectory**

Figure 13: An example of an incorrect BFS trajectory with redundant exploration to leaf nodes.

```
1   Current State: 18:[28, 23, 28, 14], Operations: []
2   Exploring Operation: 28*23=644, Resulting Numbers: [14, 644]
3   Generated Node #2: 18:[14, 644] Operation: 28*23=644
4   Current State: 18:[14, 644], Operations: ['28*23=644']
5   Exploring Operation: 644/14=46, Resulting Numbers: [46]
6   46,18 equal: Goal Reached
7   Exploring Operation: 46-28=18, Resulting Numbers: [18]
8   18,18 equal: Goal Reached
```

**Incorrect optimal path**

Figure 14: An example of an incorrect optimal path with improper handling of duplicates.

from the training target numbers and generate search trajectories using two types of heuristic-guided symbolic search algorithms:

- **DFS**: This algorithm explores nodes in increasing order of heuristic values in a depth-first manner, visiting only those with heuristic values below the target number.
- **BFS**-$b$: This algorithm explores nodes in increasing order of heuristic values in a breadth-first manner, visiting only the $b$ child nodes with the smallest heuristic values for each node. The breadth limit $b$ is chosen from 1 to 5.

We employ two types of heuristic functions in conjunction with these search algorithms:

- **Sum**: This heuristic calculates the sum of the differences between each input number and the target number.
- **Multiply**: This heuristic calculates the sum of the smallest difference between each input number and a factor of the target number.

Finally, we create 10,000 test problems for each of two different target types:

- **Seen**: These problems use the same target numbers as in the training data but with different input numbers.
- **Unseen**: These problems use the held-out target numbers for testing.

For constructing the dataset, we use the official code provided by Gandhi et al. (2024). However, we identify several critical issues in the original implementation and make the following modifications:

- The original code includes lines for exploring leaf nodes in BFS trajectories but omits them in DFS trajectories (lines 12-13 in Figure 13). Exploring leaf nodes is unnecessary because they are evaluated right after generation (lines 10-11 in Figure 13). We modify the code to eliminate this redundancy and ensure consistency with the DFS trajectories.

- The original code erroneously generates optimal paths when the input contains duplicates, removing all instances of a duplicate from the input even if only one is needed for arithmetic operations (line 2 in Figure 14). We modify the code to correctly handle duplicates.

## E.2 UNSUPERVISED PRE-TRAINING

For unsupervised pre-training, we use the official code provided by Gandhi et al. (2024). However, we find that replacing the architecture from GPT-Neo to GPT-2, while maintaining the same number of parameters, improves performance in both validation loss and test accuracy, as shown in Figure 15 and Table 3. Therefore, we choose GPT-2 over GPT-Neo as the base architecture. The configuration for GPT-2 is provided in Table 4. We use the same hyperparameter settings as in the original paper, with the specific values provided in Table 5.

Table 3: Test accuracy of unsupervised pre-trained SoS models using GPT-Neo and GPT-2.

| Architecture | Accuracy (seen) | Accuracy (unseen) |
|---|---|---|
| GPT-Neo | 0.5350 | 0.4988 |
| GPT-2 | **0.5747** | **0.5342** |

Table 4: Configuration for GPT-2.

| Attribute | Value |
|---|---|
| Embedding size | 1024 |
| The number of heads | 16 |
| The number of layers | 16 |
| Context length | 4096 |
| Attention dropout rate | 0.1 |
| Embedding dropout rate | 0.1 |
| Residual dropout rate | 0.1 |
| Data type | Bfloat16 |
| Attention implementation | FlashAttention-2 |

Table 5: Hyperparameters for unsupervised pre-training.

| Hyperparameter | Value |
|---|---|
| The number of steps | 50,000 |
| Batch size | 96 |
| Gradient accumulation steps | 1 |
| Optimizer | AdamW |
| Learning rate | 1e-5 |
| Scheduler | Cosine |
| Adam momentum | [0.9, 0.999] |
| Weight decay | 0.01 |
| Max gradient norm | 1.0 |

## E.3 SUPERVISED FINE-TUNING

For supervised fine-tuning, we also use the official code provided by Gandhi et al. (2024). We keep the same hyperparameter settings for data generation and training as in the original paper, with the

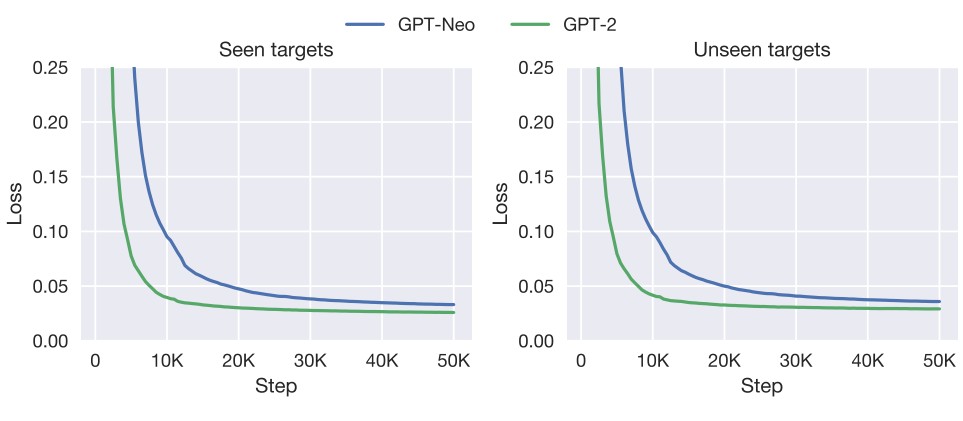

Figure 15: Validation loss of unsupervised pre-trained SoS models using GPT-Neo and GPT-2 for seen targets (**left**) and unseen targets (**right**).

specific values provided in Tables 6 and 7. However, we make a key modification to the number of problems used for data generation. In the original paper, the number of problems varies to ensure the final dataset contains 100,000 successful trajectories, leading to inconsistencies in the utilization of training data. To address this issue, we fix the number of problems to 200,000.

Table 6: Hyperparameters for supervised fine-tuning data generation.

| Hyperparameter | Value |
| --- | --- |
| The number of iterations | 3 |
| The number of problems | 200,000 |
| Temperature | 0.8 |

Table 7: Hyperparameters for supervised fine-tuning.

| Hyperparameter | Value |
| --- | --- |
| The number of steps | 20,000 |
| Batch size | 96 |
| Gradient accumulation steps | 1 |
| Optimizer | AdamW |
| Learning rate | 1e-5 |
| Scheduler | Cosine |
| Adam momentum | [0.9, 0.999] |
| Weight decay | 0.01 |
| Max gradient norm | 1.0 |

### E.4 RL FINE-TUNING

For RL fine-tuning, we implement PPO based on TRIL (Chang et al., 2023), a library that supports distributed RL training with transformers using Hugging Face Accelerate and Microsoft Deepspeed (Gugger et al., 2022; Rasley et al., 2020). However, we identify several critical issues in the original implementation and make the following modifications:

- The original code always passes a gradient accumulation step of 1 to the Deepspeed plugin, ignoring the intended setting. We modify the code to pass the correct value.
- The original code uses separate networks for the policy and value function but updates both networks simultaneously using a single optimizer. This hinders the gradient clipping, as the larger gradient overshadows the smaller gradient. We modify the code to employ separate optimizers, preventing the interference between the networks during the gradient clipping.

The hyperparameter values for RL fine-tuning are provided in Table 8. Most of the hyperparameters are derived from the default settings of TRIL, except for the learning rate and KL coefficient.

Table 8: Hyperparameters for RL fine-tuning.

| Hyperparameter | Value |
|---|---|
| The number of epochs | 200 |
| The number of rollouts per epoch | 128 |
| Temperature | 1.0 |
| The number of PPO epochs | 4 |
| Batch size | 32 |
| Gradient accumulation steps | 4 |
| Discount factor | 1.0 |
| GAE lambda | 0.95 |
| Advantage whitening | False |
| Optimizer | AdamW |
| Learning rate | 1e-7 |
| Scheduler | Constant |
| Adam momentum | [0.9, 0.999] |
| Weight decay | 0.01 |
| Max gradient norm | 1.0 |
| KL controller | Constant |
| KL coefficient | 0.01 |

### E.5 COMPUTATIONAL RESOURCES

We conduct all experiments on an internal HPC cluster, with each node consisting of 2 AMD EPYC 7402 CPUs and 750GB of RAM, using the PyTorch deep learning framework (Paszke et al., 2019). For supervised pre-training and fine-tuning, we utilize 4 NVIDIA A100 GPUs (80GB VRAM each). For RL fine-tuning, we utilize 4 NVIDIA RTX 3090 GPUs (24GB VRAM each). For inference, we utilize a single NVIDIA RTX 3090 GPU.

## F COMPARISON OF ITERATIVE APA WITH PPO

Gandhi et al. (2024) adopt iterative APA for RL fine-tuning instead of PPO. In this approach, a policy is fine-tuned using APA over multiple iterations, with the best-performing policy from each iteration serving as the reference for the subsequent iteration. However, this method can be cumbersome, as it requires periodically saving and evaluating checkpoints to determine the best-performing policy.

We observe that a single iteration of PPO achieves better performance than iterative APA with fewer rollouts, as shown in Table 9. Notably, SoS+PPO in the operation-level MDP outperforms iterative APA, with an accuracy gain of 4% for both seen and unseen targets. For a fair comparison, we train the SoS model using the same dataset and architecture as in the original paper. Therefore, we choose operation-level PPO over iterative APA as the base RL algorithm.

Table 9: Test accuracy of RL fine-tuned models using GPT-Neo. The results of SoS+Iterative APA are replicated from the original paper.

| Model | # rollouts | Accuracy (seen) | Accuracy (unseen) |
|---|---|---|---|
| SoS | - | 0.5040 | 0.4844 |
| SoS+Iterative APA | 64,000 | 0.5652 | 0.5423 |
| SoS+PPO (token) | 25,600 | 0.5626 | 0.5467 |
| SoS+PPO (operation) | 25,600 | **0.6015** | **0.5984** |

