# OpenReview forum: "Guided Stream of Search: Learning to Better Search with Language Models via Optimal Path Guidance"
_ICLR.cc/2025/Conference — ICLR 2025 Conference Withdrawn Submission_

### Official Review · Reviewer_1yZo · 2024-10-21

**Soundness:** 3
**Presentation:** 3
**Contribution:** 2
**Rating:** 3
**Confidence:** 3

**Summary:**

This paper introduced the Guided Streat of Search, which integrates optimal solutions into the self-generation process of LLMs to improve their search and planning capabilities. The main contribution is extending the existing Stream of Search (SoS) approach to Guided Stream of Search (GSoS), which incorporates optimal solutions into the self-generation process in a progressive manner. GSoS uses unsuccessful search trajectories as contexts to integrate each intermediate action from the optimal solution, producing high-quality search trajectories that are then used for SFT. GSoS is evaluated on a search benchmark and demonstrates outperformance in comparison to both SFT and RLHF baselines, regarding both seen targets and unseen targets.

**Strengths:**

[+] The overall presentation and structure are well-organized. The introduction, preliminary, and method sections are well-written. The threads are easy to follow.

[+] The results and analysis of the experiments are detailed and comprehensive. The authors provide extensive experiment results and analyze them in detail. In my opinion, this paper is fine with its empirical results and analysis.

[+] All the codes and hyperparameters are open-sourced for reproducibility.

[+] I believe this method has potential applications for larger problems and more advanced models. Augmenting search and planning trajectories could be a crucial step in training models like o1.

**Weaknesses:**

[-] The evaluation benchmark is not convincing to me. It appears that this benchmark can easily be formulated as a real search problem, making the use of an LLM unnecessary. I think the authors should consider testing their framework on a more complex benchmark.

[-] It's doubtful that the unseen targets in Countdown can be considered a valid evaluation of generalization, given the high similarity between the supposedly different tasks in the dataset.

[-] The backbone model, GPT-2, is somewhat outdated. Additionally, I could not find an explanation provided for choosing GPT-2 over other models.

On a minor note, I do not observe any planning capability (i.e., the ability to plan ahead of actions) from this method or within the benchmark, despite its repeated emphasis in the paper.

**Questions:**

1. How does the performance of GSoS compare with other state-of-the-art algorithms in terms of search and planning capabilities? Can leading search and planning algorithms be transferred to this benchmark and be evaluated?

2. What's the reason behind choosing GPT-2 as the backbone model? Is it possible to replicate the experiments with more advanced open-sourced models?

---

### Official Review · Reviewer_Rj6q · 2024-11-01

**Soundness:** 2
**Presentation:** 2
**Contribution:** 2
**Rating:** 5
**Confidence:** 3

**Summary:**

This work explores how to leverage optimal solutions to enhance the search and planning abilities of language models. The authors propose guided stream of search (GSoS), which seamlessly incorporates optimal solutions into the self-generation process in a progressive manner, producing high-quality search trajectories for training. GSoS can significantly enhance the search and planning abilities of language models on Countdown, a simple yet challenging mathematical reasoning task.

**Strengths:**

1. The paper is well-written and easy to follow

2. The proposed method is simple and intuitive

3. Good experimental results and detailed analysis on Countdown benchmark

**Weaknesses:**

1. The experiments are only conducted on one single benchmark. There are many other datasets requiring complex reasoning. At least one of them, such as LogiQA2, should be investigated.

2. The authors use a 250M model for experiments which is quite small. For complex planning and reasoning, larger language models should be considered.

3. How about the comparison to this simple baseline? For the given query, we sample plenty of trajectories from the model and construct a DAG using the sampled trajectories. Then we can sample different types of search paths from the DAG for training.

**Questions:**

See weaknesses

---

### Official Review · Reviewer_NMuC · 2024-11-04

**Soundness:** 3
**Presentation:** 3
**Contribution:** 3
**Rating:** 5
**Confidence:** 4

**Summary:**

The authors introduce GSoS, a method to improve the planning and reasoning capabilities of language models by integrating optimal solutions within search processes. Unlike prior approaches that rely solely on self-generated, often suboptimal search trajectories, GSoS incorporates optimal solutions progressively, guiding the model toward more structured search trajectories. These trajectories are distilled through SFT, which, combined with subsequent RL training, enhances performance on the planning task Countdown.

**Strengths:**

1. The paper presents a simple and intuitive approach for improving planning tasks in LLMs by incorporating optimal solutions into trajectory generation process, which enhances the quality of generated trajectories and overall training outcomes.
2. The paper is clearly written, making the proposed method and experiment findings accessible.

**Weaknesses:**

1. A key baseline—using SFT with the optimal solutions (BC)—is missing. While the authors discuss BC's limitations on unseen tasks, including it in the evaluation would provide a more comprehensive comparison, especially since the main contribution of this approach is incorporating optimal solutions into the data construction process.
2. The proposed approach is only validated on a single test bed, Countdown, which may leave readers questioning its generalizability to other planning tasks. Including an additional test bed, such as those from Beyond a* [1], would strengthen the paper’s claims, particularly as this work builds on and seeks to improve upon SoS (Gandhi et al., 2024).

**Minor Issue:**
- Line 111: The purpose of transforming $x$ through a series of operations to obtain $\\hat{y}$ is unclear, as $x$ already contains both input and output states?


[1] Beyond a*: Better planning with transformers via search dynamics bootstrapping.

**Questions:**

Please refer to weaknesses.

---

### Official Review · Reviewer_cqn2 · 2024-11-07

**Soundness:** 2
**Presentation:** 3
**Contribution:** 2
**Rating:** 3
**Confidence:** 4

**Summary:**

This paper proposes Guided Stream of Search (GSoS), a novel method that combining the optimal path as well as the search trajectories of a searching scenario into a sequence, which is used as the training data instance for LLMs to acquire better planning and search performances. The authors have conducted experiments on Countdown, a mathematical reasoning benchmark with branching factor in square complexity of the inputs at each searching step. The experimental results demonstrated the effectiveness of GSoS, especially with RL that functions on the operation level.

**Strengths:**

- This paper studies complex reasoning and planning of LLMs, which is an important topic in LLM research.
- The idea of integrating more exploratory trajectory segments into the context of the optimal subgoal makes sense, as it steers LLMs to learn to pivot to the optimal path.
- Setting up the RL training on the operation level effectively accelerates the learning process, which is supported by comparison experiments.

**Weaknesses:**

Please refer to the Questions listed below.

**Questions:**

- When conditioning the optimal path with a partial exploration path, is it equivalent to a self-reflection process (and the the self-reflection succeeds with one reflection trial)? If so, what is the novelty of GSoS over a RL reflection-tuning method, or RL finetuning with Chain-of-Hindsight [1]?
  - Furthermore, have the authors tried to compile the trajectories by exploring more than one non-subgoal node in advance of the subgoal node, and ablate the effect with those containing only one non-subgoal node ahead of each corresponding subgoal node?

  [1] Liu et al., Chain of Hindsight Aligns Language Models with Feedback. ICLR 2024.

- The effectiveness of GSoS is only demonstrated on one benchmark. The proposed method should be benchmarked on more scenarios to demonstrate its superiority.

- In Lines 192-194, it is claimed that "Fine-tuning on these trajectories may lead to significant changes in the model’s weights, potentially degrading its search and planning abilities. Therefore, it is crucial to explore methods for effectively integrating optimal solutions to produce trajectories that maintain both high likelihood and quality." It would be beneficial if the authors provide more experimental supports for why direct finetuning leads to the degradation of the search and planning abilities. Specifically, if it is supported by the main experiments where GSoS outperforms SoS, additional qualitative analysis and case studies are needed for the direct comparison between GSoS and SoS, and it would be helpful to provide cases where GSoS+finetuning succeeds while SoS+finetuning fails.

- In Lines 306-307, it is demonstrated that "even when multi-step returns with GAE are used for training the value function." It would be beneficial if the authors could show the experiments that verify this claim.

- In Line 5 of Algorithm 2: what is M(y|x)?

---

### Note · Authors · 2025-01-02

I have read and agree with the venue's withdrawal policy on behalf of myself and my co-authors.